# Genetic Factors That Could Affect Concussion Risk in Elite Rugby

**DOI:** 10.3390/sports9020019

**Published:** 2021-01-22

**Authors:** Mark R. Antrobus, Jon Brazier, Georgina K. Stebbings, Stephen H. Day, Shane M. Heffernan, Liam P. Kilduff, Robert M. Erskine, Alun G. Williams

**Affiliations:** 1Sports Genomics Laboratory, Department of Sport and Exercise Sciences, Manchester Metropolitan University, Manchester M1 5GD, UK; j.brazier2@herts.ac.uk (J.B.); g.stebbings@mmu.ac.uk (G.K.S.); a.g.williams@mmu.ac.uk (A.G.W.); 2Sport and Exercise Science, University of Northampton, Northampton NN1 5PH, UK; 3Department of Psychology and Sports Sciences, University of Hertfordshire, Hatfield AL10 9AB, UK; 4Faculty of Science and Engineering, University of Wolverhampton, Wolverhampton WV1 1LY, UK; stephen.day@wlv.ac.uk; 5Applied Sports, Technology, Exercise and Medicine (A-STEM) Research Centre, College of Engineering, Swansea University, Swansea SA1 8EN, UK; s.m.heffernan@swansea.ac.uk (S.M.H.); l.kilduff@swansea.ac.uk (L.P.K.); 6Research Institute for Sport & Exercise Sciences, Liverpool John Moores University, Liverpool L3 3AF, UK; r.m.erskine@ljmu.ac.uk; 7Institute of Sport, Exercise and Health, University College London, London WC1E 6BT, UK

**Keywords:** genomics, rugby, polymorphisms, concussion, mild traumatic brain injury

## Abstract

Elite rugby league and union have some of the highest reported rates of concussion (mild traumatic brain injury) in professional sport due in part to their full-contact high-velocity collision-based nature. Currently, concussions are the most commonly reported match injury during the tackle for both the ball carrier and the tackler (8–28 concussions per 1000 player match hours) and reports exist of reduced cognitive function and long-term health consequences that can end a playing career and produce continued ill health. Concussion is a complex phenotype, influenced by environmental factors and an individual’s genetic predisposition. This article reviews concussion incidence within elite rugby and addresses the biomechanics and pathophysiology of concussion and how genetic predisposition may influence incidence, severity and outcome. Associations have been reported between a variety of genetic variants and traumatic brain injury. However, little effort has been devoted to the study of genetic associations with concussion within elite rugby players. Due to a growing understanding of the molecular characteristics underpinning the pathophysiology of concussion, investigating genetic variation within elite rugby is a viable and worthy proposition. Therefore, we propose from this review that several genetic variants within or near candidate genes of interest, namely *APOE*, *MAPT*, *IL6R*, *COMT*, *SLC6A4*, *5-HTTLPR*, *DRD2*, *DRD4*, *ANKK1*, *BDNF* and *GRIN2A*, warrant further study within elite rugby and other sports involving high-velocity collisions.

## 1. Introduction

Rugby union (RU) and rugby league (RL) are both full-contact collision-based codes of rugby, which have some of the highest reports of concussion in professional sports (“rugby” will be used to refer to both RU and RL). Rugby-related concussions have been the focus of recent concern over the potential short- and long-term neurodegenerative consequences. In addition, athletes who have had a prior concussion have a higher risk of repeated concussions and subsequent time-loss injury [1,2,3,4]. There is a reported increased risk of potential short- and long-term consequences associated with concussion such as increased injury risk, cognitive impairment, forms of dementia, chronic post-concussion syndrome, migraines, sleep dysfunction, anxiety, post-traumatic stress disorder and second-impact syndrome [5,6,7,8,9,10,11,12,13,14]. These consequences could interrupt or terminate an athletic career, causing short- or long-term ill health.

Sport-related concussion has been defined as a traumatic brain injury (TBI) induced by biomechanical forces [5]. However, many factors contribute to concussion risk such as age, sex, playing position, playing level, behaviour, rules of the sport, neck strength, nutrition, and sleep quality [15,16,17]. Concussion has been widely studied in relation to environmental factors, especially in rugby, where factors considered include activity when concussion occurred (e.g., tackling/being tackled), playing experience, history of concussion, positional differences, use of protective equipment (e.g., headgear/mouth guards) and return-to-play protocols and standard of competition [18,19]. However, a further step to better understanding inter-individual variability involves genetic variation and its association with concussion and related phenotypes. Evidence already exists suggesting an association between several genetic factors and inter-individual variability in traumatic brain injury incidence and severity [20,21,22,23,24,25,26,27,28].

Classical genetic studies (twin or family studies) quantify the heritability of phenotypic traits [27]. As concussion is only experienced by a small proportion of the population [28], recruiting a sufficient number of twins/family members who have experienced concussion is difficult (though not impossible) and has not been undertaken, to our knowledge. Consequently, a classical study on the inheritance of concussion risk, to elucidate the relative contribution of environmental versus genetic factors affecting inter-individual variability in concussion incidence, severity and outcome, would be extremely valuable. Many other sport-related injuries or risk factors for injury have substantial genetic contributions to their inter-individual variability, such as tennis elbow (epicondylitis), for which heritability has been estimated at a substantial ~40% in women [29] and bone mineral density (a predictor of osteoporotic fracture), for which heritability is even greater at 50–85% [30]. Substantial heritability estimates for brain structure (~90%) and cognitive performance (~60%) have also been reported [31,32,33,34]. Given these and other observations of substantial genetic contributions to inter-individual variability in most human traits, it is likely that a substantial genetic component also applies to concussion. 

Indeed, the substantial inter-individual variability in injury occurrence, and in outcomes following concussion, is probably due to the interaction of multiple genes in a polygenic manner that reflects the complex pathophysiology [35,36]. Prediction of recovery and future risk is therefore currently difficult [5]. This unexplained inter-individual variability could suggest a future role for genetic screening of concussion-associated risk polymorphisms in order to (i) stratify potential risk of initial injury, for individuals (ii) identify players with a greater risk of prolonged recovery and potential concussion-associated neurological issues, (iii) identify those at risk of repeated concussions, (iv) provide further insight into concussion pathophysiology, and (v) inform concussion management strategies at a practical level in elite sport.

Therefore, the aims of this narrative review are to (1) describe the current data on incidence rates and severity of concussion in elite rugby; (2) provide an overview of the mechanisms and pathophysiology of concussion; (3) evaluate how genetic variation could affect predisposition for and recovery from concussion; and (4) inform the future direction research regarding genetic aspects of concussion in rugby.

## 2. Incidence Rate and Severity of Concussion in Rugby

The professionalisation of rugby has resulted in alterations in the physical characteristics of players [37,38,39,40]. These alterations in physical characteristics such as body mass, strength, power and speed have increased the physical demands of modern rugby, such as more tackles and rucks per match [40,41,42,43,44]. This increased physicality has contributed to increased incidence rates of concussion in rugby [45,46].

There are many similarities in anthropometric and physiological characteristics of players in RU and RL that reflect comparable physical demands including frequent, heavy physical contact in both rugby codes [40]. Elite rugby (RU and RL) has been reported to have a concussion incidence of ~8–28 concussions per 1000 match hours [47,48], which is lower than sports such as horse racing (17–95) and boxing (13) but higher than sports such as soccer (0.4) [49,50,51]. Seventy percent of head injury assessments in elite RU as a result of a tackle are experienced by the tackler and 30% by the ball carrier [52]. This concussion risk is influenced by athlete speed, playing position, impacting force, body position, type of tackle, tackle technique, and physiological and anthropometric characteristics [53,54].

Recovery from concussion has been defined as a return to sport that encompasses a resolution of post-concussion-related symptoms and a return to clinically normal balance and cognitive functioning [5]. Within 7–10 days, 80–90% of adults with sport-related concussions could be clinically recovered and returned to play (Figure 1) [5,55,56].

For 10–20% of concussion cases, symptoms can persist for >10 days [55]. Time taken to recover from a concussion differs for individuals, as 6.5% of concussed athletes have been reported to not return to play until 14 days post-concussion. For 1.6% of concussed athletes, recovery can take longer than 14 days and these individuals could have chronic post-concussion symptoms for up to 12 months [56,57].

Concussion prevalence during the Rugby World Cups has seen a small increase from ~14% of all injuries in 2015 to ~16% in 2019 [58,59]. In the English Rugby Premiership (the top tier of competition in England), concussion incidence increased dramatically from 8 per 1000 match hours in the 2013–2014 season to 22 in 2016–2017, although this is thought to be largely due to increased awareness and reporting [60]. However, concussion incidence within the English RU Premiership decreased to 18 concussions per 1000 match hours in 2017–2018 (~1 concussion per match) [48]. In elite RL, concussion incidence in the National Rugby League (the top tier of competition in Australia) has ranged from ~9 to 28 concussions per 1000 player match hours over a 17 year period with a tendency to increase over time [61,62,63].

The incidence of concussions in RU is similar for forwards (4–19 per 1000 player match hours) and backs (5–18 per 1000 player match hours) [18,64]. In RL, incidence of concussions ranges from 12 to 48 per 1000 player match hours in forwards and a similar 14 to 44 per 1000 player match hours in backs [65]. Concussion incidence in both codes during training is much lower, accounting for only ~5% of concussions (0.03–0.07 per 1000 player training hours) [18,66]. Fluctuations in incidence over time could be attributed to developments in concussion education or operational strategies such as using ‘Hawkeye’ video analysis [48]. Increased awareness of players, support staff and coaches could account for the increased incidence of concussion reported in recent years [48]. Awareness is thought to be increased due to education initiatives by rugby governing bodies and player associations involving increased recent media attention [67].

The average range of concussion severity in RU ranges from 9 to 21 days absence (period from injury to availability for match selection) [4,48,61,65]. However, inter-individual variability means that severity can range from 2 days to >84 days absence [49]. Data from the 2013–2015 Super League RL seasons suggest severity can range from 9 to 15 days absence [47].

## 3. Mechanisms of Concussion

Rugby-related concussions can be the result of either direct head contact or inertial causes, but each concussion is a unique event. Contact injuries (e.g., from collisions) cause the brain to impact on the internal surfaces of the skull. Particularly injurious are incidents involving the frontal and temporal fossae regions due to ridges and bony protuberances that deform brain tissue [68]. Kinematic analysis indicates that inertial forces from direct or indirect impacts resulting in angular/linear acceleration/deceleration of the brain from head and neck motions can lead to concussion [69].

The contributions of angular or linear acceleration/deceleration to concussion is debated in the literature [70]. Linear acceleration is associated with changes in pressure gradients within the skull, compared to angular acceleration/deceleration that is associated with shear stresses on the brain forcing tissues to slide over one another and stretch [71]. Shear and stretch mechanical forces stretch axons to the point of axotomy (physical breaking) or partial breaking in areas, such as grey and white matter junctions, small blood vessels and axonal projections [69,72,73].

Concussions appear to vary in impact locations (front, top, back and sides of the head), linear acceleration/deceleration magnitude (61–169 g in collegiate American Football players, although there are concerns about the validity of those high values [74]) and clinical outcomes [75]. However, head impacts from high-magnitude angular acceleration/deceleration result in more severe clinical outcomes due to the propensity of brain tissue to deform more readily from shear forces and are the predominant mechanism in multifocal concussion [71,75]. A tackle or collision may produce whiplash, which in turn produces both linear and angular acceleration/deceleration to the player’s brain [75].

## 4. Pathophysiology of Concussion

In rugby, the primary mechanical stress injury to neurons is likely the result of a collision that elicits a neuronal stretch. A stretch of ~10–20% of a neuron’s resting length within 100 ms (sublethal axonal injury threshold) can trigger the secondary biochemical response of the neurometabolic cascade [76,77]. The resultant microstructural damage caused by the stretch is hypothesised to be the root cause of all forms of TBI [78,79,80]. The neurometabolic cascade following a concussive event (Figure 2) has been reviewed by Giza and Hovda [76,77].

The initial disturbance and stretch result in the release of depolarising extracellular K^+^ due to voltage-dependent channels opening in the neuronal membranes and this can last up to 6 h post-concussion [81,82]. Further K^+^ flux is caused by the release of the excitatory amino acid glutamate [83]. Proteolytic digestion of the axon membrane skeleton occurs due to Ca^2+^ activation of cysteine proteases and apoptotic genetic signals [84]. Ca^2+^ influx has been reported to contribute to axonal microtubule breakdown 6–24 h after a concussive event [82]. During smaller insults to the brain, surrounding glial cells remove extracellular K^+^ in order to maintain homeostasis [85]. However, this cannot be achieved during larger concussive events and greater quantities of excitatory amino acids are released, resulting in ‘spreading depression’ [86]. Multiple mechanisms are responsible for elevated Ca^2+^ levels—firstly, the physical disruption of membranes through primary injury [87]; secondly, increased glutamate binds receptors such as n-methyl-d aspartic acid (NMDA) subunit NR2A, increasing Ca^2+^ influx through the NMDA channel, prolonging neuronal dysfunction [88].

Disruption of ionic homeostasis leads to an energy crisis within the injured brain. Re-establishment of ionic homeostasis is further attempted by the employment of ATP-fuelled membrane pumps, which results in increased glycolysis to meet energy requirements due to reduced activity of cerebral oxidative metabolism and reduced cerebral blood flow of up to 50% [89]. Increased intracellular Ca^2+^, Na^+^ and K^+^ can result in swelling and contribute to further reduced cerebral blood flow [90]. Mitochondrial oxidative metabolism is impaired due to the influx of extracellular Ca^2+^, thus contributing to the energy crisis [91]. As part of the neurometabolic cascade, pro- and anti-inflammatory cytokines are released [92]. Cytokines from this neuroimmune response can play both beneficial and detrimental roles in the neuroinflammatory response following a concussion [92].

## 5. Genetic Associations with Concussion

Genome-wide association studies (GWAS) enable the genome to be searched for unsuspected variations as opposed to candidate areas as in a gene association study [93,94]. In elite sport, however, the maximum number of individuals available for study is limited. For example, the English Rugby Premiership comprises ~600 players and Super League ~360 players. This limited sample size reduces the feasibility of GWAS, as considerably larger sample sizes are often required to meet the traditionally accepted significance value of *p* < 5 × 10^−8^. Genetic association studies utilising a candidate gene approach enable the study of genetic variance within a complex polygenic trait [95]. An advantage of the candidate gene approach is that genes are selected utilising an a priori hypothesis based on the biological function of a particular protein and the specific phenotype [95,96], and statistical power can be sufficient to test specific hypotheses using sample sizes available in elite sport. A disadvantage of the candidate gene approach is that only genes/variants already suspected are investigated, excluding the possibility of discovering hitherto unsuspected genes/variants that might be important.

Functionally, significant polymorphisms (single-nucleotide polymorphisms (SNPs), repeat polymorphisms, insertions or deletions) used in the candidate gene approach are often selected based on the likeliness to affect gene function. Priority polymorphisms include those that alter an amino acid in a protein (missense variation) or produce a stop codon (nonsense variation) [95]. Polymorphisms in promoter and regulatory regions of a gene could also have functional consequences by influencing transcription rate [95].

### 5.1. Candidate Genetic Variants

A complex array of physiological and psychological responses to concussion have been reported, so the proposed influencing genes have been categorised into four groups. These groups are based on current knowledge and some genes fit into more than one category due to the nature of their functions: 1. genes that affect the severity of concussion; 2. genes that affect repair and plasticity of the brain; 3. genes that affect post-concussion cognitive behavioural capacity; and 4. genes that affect personality traits and concussion risk. The genes are listed in Table 1 and addressed in Section 5.1.1, Section 5.1.2, Section 5.1.3, Section 5.1.4, Section 5.1.5, Section 5.1.6, Section 5.1.7, Section 5.1.8, Section 5.1.9, Section 5.1.10, Section 5.1.11, Section 5.1.12, Section 5.1.13, Section 5.1.14, Section 5.1.15 and Section 5.1.16.

#### 5.1.1. Apolipoprotein E

*Apolipoprotein E* (*APOE*) is the most researched gene in respect to TBI. APOE isoforms have both protective and detrimental effects (Appendix A). These effects are dependent upon which specific alleles an individual carries and thus gene expression after the TBI event. *APOE* has three common allelic isoforms ε2, ε3 and ε4 which differ by amino acid substitutions at residues 112 and 158 [148]. Two C/T SNPs at residues 112 (rs429358) and 158 (rs7412) result in amino acid substitutions of arginine (C) to cysteine (T) at each residue (Appendix A). The two nonsynonymous SNPs at residues 112 and 158 can produce the three isoforms of ε2, ε3, ε4 and six possible genotypes (Table 2) of relevance to concussion.

APOE isoforms have differing effects on neurite extension, which can influence ability to recover post-concussion. APOE ε3 stimulates neurite growth in cultured neuronal cells [97,98]. In contrast, APOE ε4 suppresses neurite growth [97,98]. These findings suggest that *APOE* ε2 and ε3 would provide more effective neuronal repair, such as proliferation of dendrites post-concussion compared to *APOE* ε4 [97,98]. In addition, the ε4 alleles have been associated with the formation of neurodegenerative amyloid plaques (Aβ) and increased risk of Alzheimer’s disease [99].

Despite the pathophysiological roles that APOE ε4 plays in TBI, studies associating *APOE* ε4 and sport-related concussion are few and findings are conflicting. Kristman et al. [100] showed no association between *APOE* ε4 carriers and incidence of concussion in Varsity level athletes. These findings have been supported by Terrell et al. [2] and Tierney et al. [1], who also reported no association between concussion incidence and *APOE* genotypes in collegiate athletes. More recently, Abrahams et al. [149] reported no association in *APOE* ε2, ε3 and ε4 genotypes and incidence of concussion in a mixed cohort of youth, amateur and professional South African RU players.

Early findings from Jordan et al. [150] indicated that *APOE* ε4 carrier boxers experiencing high-exposures (>12 professional bouts) had greater chronic brain injury scale scores than non-ε4 carrier high-exposure boxers. Indeed, it has been suggested that the *APOE* ε4 allele may be responsible for up to 64% of the ‘hazardous influence’ of TBI [151] and athletes who possess the ε4 allele suffer from prolonged physical (Cohen’s *d’* = 0.87) and cognitive (*d’* = 0.60) symptomatic responses to concussion [152].

Polymorphisms within the promoter region of *APOE* have been associated with functional regulation of *APOE* transcription and quantitative impacts on apolipoprotein E levels in brain tissue, as well as unfavourable outcomes post-TBI [101,102]. It has been hypothesised that the -219 T allele at rs405509 exacerbates the effects of the ε4 allele through upregulation of APOE gene transcription and increased Aβ plaque accumulation [102].

Lendon et al. [102] observed an association between individuals with rs405509 TT genotype and unfavourable outcomes post-TBI over a six-month recovery period. Tierney et al. [1] reported that carriers of the T allele had an 8-fold greater risk of experiencing two or more concussions. Similarly, Terrell et al. [2] suggest that the TT genotype is associated with a 3-fold greater risk of previous concussion and a 4-fold greater risk of a history of concussion with loss of consciousness. In contrast, Abrahams et al. [149] reported that TT genotype was associated with a 45% reduced risk of and concussion and the T allele was associated with a <1-week recovery period post-concussion in a mixed cohort of youth and professional South African RU players. These conflicting findings could be in part due to differences in sport and, in particular, geographic ancestry of the participants. Nevertheless, the plausible physiological mechanisms and the limited number of association studies warrant further investigation of this concussion-associated SNP.

#### 5.1.2. Microtubule-Associated Protein Tau Polymorphisms

The functions of *microtubule-associated protein tau* (*MAPT*) include encoding the tau protein that modulates microtubule formation, structural stabilisation of the neuronal axons and driving growth of neurites [103,104]. Elevated post-TBI plasma levels of tau have been observed for up to 90 days [153]. Autopsies on American football players’ brains who had experienced repetitive concussions indicate the presence of neurofibrillary tangles (aggregates of hyperphosphorylated tau protein) and neuropil filaments (abnormal neurite formations) [154]. These neurotoxic formations have been associated with neurodegenerative diseases such as Alzheimer’s disease, chronic traumatic encephalopathy, Parkinson’s disease, frontotemporal dementia and a range of other neurodegenerative diseases under the term tauopathies [155,156,157,158]. The *MAPT* (rs10445337) T/C SNP is postulated to modulate the formation of neurotoxic-paired helical filaments composed of hyperphosphorylated tau [159,160] (Appendix A).

Terrell et al. [2] reported a nonsignificant observation that the *MAPT* rs10445337 TT genotype was weakly associated with a history of one or more concussions (odds ratio, 2.1; 95% CI, 0.3 to 14.5). Similarly, in a later study, no association was observed between concussion incidence and *MAPT* rs10445337 [22]. Recently, other MAPT SNPs (rs2435211 and rs2435200) have been implicated as potential pathophysiological mechanisms in RU players [20]. The AG genotype of rs2435200 has been associated with an increased risk of sustaining multiple concussions in senior (>18 years old) RU players [20]. In addition, the T-G haplotype (rs2435211 and rs2435200) has been associated with an increased risk of sustaining a concussion in senior amateur and elite RU players [20].

#### 5.1.3. Neurofilament Heavy Polymorphism

Approximately 50% of the neuronal cytoskeleton is comprised of light, medium and heavy neurofilaments [105]. A function of the neuronal cytoskeleton is to resist the resultant strain caused by biomechanical forces during a head impact [105]. In one study, a small cohort of 48 college level athletes with self-reported history of concussion were genotyped for an A/C polymorphism (rs165602) of the *neurofilament heavy (NEFH)* gene (Appendix A) [161]. The authors observed no association between the polymorphism and incidence or severity of concussion in college athletes.

#### 5.1.4. Membrane Metalloendopeptidase Polymorphism

The *membrane metalloendopeptidase* (*MME*) gene encodes the neprilysin protease (Appendix A) that degrades amyloid plaque (Aβ) proteins [106]. A GT repeat within the promoter region of *MME* regulates expression of neprilysin in neurons [107]. Greater Aβ deposits were observed after severe TBI in patients who had long *MME* GT repeats (>41) [108]. It was also observed that carrying at least one 22-repeat allele was associated with increased risk of Aβ plaque deposition and carrying at least one 20-repeat allele associated with decreased risk.

#### 5.1.5. Brain-Derived Neurotrophic Factor Polymorphism

*Brain-derived neurotrophic factor* (*BDNF*) is a gene that affects the repair and plasticity of neurons. It is a member of the neurotrophin family, responsible for mediating neuronal plasticity [109,110]. Neurotrophins aid in the development, differentiation, proliferation and survival of neurons (dopaminergic, serotonergic and cholinergic) [109,111]. A widely studied SNP is the C to T missense variation at nucleotide 196 resulting in a valine to methionine (Val66Met) substitution at codon 66 [162] (Appendix A). BDNF mRNA is upregulated post-TBI event and can remain elevated for up to three days post-TBI [162,163,164,165]. *BDNF* plays an important role in strengthening existing synaptic connections and modulating the creation of new synapses [110]. The Met allele impairs intracellular tracking and packaging of precursor-BDNF (pro-BDNF) and activity-dependent secretion of BDNF [162].

Dretsch et al. [166] reported that ~17% of Met/Met homozygotes suffered a concussion during military deployment compared to ~4% of Val carriers. Narayanan et al. [167] found that the rs6265 polymorphism was associated with neurocognitive performance in concussed individuals acutely and 6 months post-event, as Val/Val homozygotes performed better in measures of memory, executive function, attention and overall cognitive performance [167].

#### 5.1.6. Glutamate Ionotropic Receptor NMDA Type Subunit 2A Variant

*Glutamate ionotropic receptor NMDA type subunit 2A* (*GRIN2A*) encodes glutamate-gated ion channel proteins. A variable number tandem repeat (VNTR) polymorphism within the promoter region of *GRIN2A* modulates n-methyl-d aspartic acid (NMDA) receptors within the brain. The NMDA NR2A subunit has been associated with neuronal plasticity, spatial and episodic memory [112,113]. The VNTR GT (rs3219790) repeat within the promoter region affects transcriptional activity in a length-dependent manner (Appendix A) [114,115]. The longer the GT repeat, the lower the *GRIN2A* promoter activity [115]. Longer repeats of >25 (GT) can be termed long alleles (L) and shorter repeats of <25 (GT) termed short alleles (S) [114,115].

Findings from McDevitt et al. [168] indicate that L allele carriers were twice as likely to recover in >60 days than S allele carriers. A dose response was also reported: LL carriers were 6-fold more likely to have a prolonged recovery (>60 days) compared to individuals of SS genotype.

#### 5.1.7. Catechol-O-methyltransferase Polymorphism

The *catechol-O-methyltransferase* (*COMT*) gene has been postulated to affect post-concussion cognitive behavioural capacity [116]. *COMT* encodes an enzyme that methylates and in turn deactivates catechol-based neurotransmitters such as synaptic dopamine and noradrenaline [117] (Appendix A). Optimal cognitive function is affected by the prefrontal cortex’s sensitivity to dopamine, which makes *COMT* an ideal candidate gene for influencing inter-individual variability in cognitive function post-concussion. A widely studied SNP within the *COMT* gene is the G to A missense variation at codon 158 resulting in a valine (Val) to methionine (Met) amino acid substitution. Val/Val carriers have greater COMT activity than Met/Met carriers [118].

Lipsky et al. [116] reported that Val allele carriers performed poorer on tests of executive function compared to Met allele carriers post-TBI. More recently and in contrast, Willmott et al. [169] reported no significant influence of *COMT* polymorphisms on cognitive performance in moderate to severe TBI patients. However, Lipsky et al. [116] employed a battery of executive function tests including the Wisconsin Card Sorting Test, while Willmott et al. [169] used the Glasgow Outcome Scale-Extended as a measure of functional outcome post-TBI. Mc Fie et al. [21] reported that Met carriers in a cohort of youth and professional South African RU players were ~3-fold more likely to have a history of concussion and, accordingly, it has been postulated that elevated dopamine could increase impulsivity and risk taking meaning Met allele carriers could place themselves at increased risk of sustaining a concussion [170,171].

#### 5.1.8. Ankyrin Repeat and Kinase Domain Containing 1 Polymorphism

*Ankyrin repeat and kinase domain containing 1* (*ANKK1*) is a dopaminergic gene known to affect working memory, reward and motivation [119,120]. *ANKK1* was originally referred to as *Taq1A* and is in linkage disequilibrium (D’ > 0.80) with the 10 kB downstream *dopamine receptor D2* (*DRD2*) gene [121]. The *ANKK1* C/T (rs1800497) SNP is hypothesised to be in a regulatory region within *DRD2* (Appendix A) [121]. ANKK1 is expressed in astroglial cells (a type of brain-derived glial cell), post-mitotic neurons and neural precursors from neurogenic niches and as a member of the serine/threonine receptor-interacting protein kinases is responsible for dopaminergic signal transduction and cellular response [121,122].

*ANKK1* polymorphisms affect dopamine transporter densities within the striatum which influences working memory, reward and motivation [121,122]. The T allele of *ANKK1* has been associated with a 30–40% reduction in the expression of D2 receptors within the ventral striatum [123,124]. *ANKK1*’s polymorphic role in modulating working memory and cognitive performance vis-à-vis concussion/TBI is limited to three studies. McAllister et al. [172,173] observed concussed T allele carriers performed significantly worse in measures of learning, working memory and response latencies. Similarly, Yue et al.’s [174] findings support McAllister et al. [172,173] and indicate a dose-dependent association with the T allele. Thus, this polymorphism could influence recovery from a concussive event.

#### 5.1.9. Dopamine Receptor-Related Polymorphisms

Dopamine receptors (*DRD2* and *DRD4*) have been associated with risk-taking behaviours (impulsivity, behavioural inhibition and novelty seeking) [125,126]. Polymorphisms within *DRD2* and *DRD4* genes have been postulated to affect personality traits, possibly via inhibition of neurotransmission [175]. *DRD2* SNPs rs12364283 (A/G) and rs1076560 (C/A) have been associated with altered D2 receptor expression (Appendix A) [175]. The *DRD4* promoter rs1800955 C allele has been associated with higher DRD4 expression compared to the T allele (Appendix A) [125]. Furthermore, the *DRD4* (rs1800955) CC genotype and inferred haplotype of *DRD2* (rs12364283–rs1076560)–*DRD4* (rs1800955) A–C–C alleles associated with decreased concussion susceptibility in junior South African RU players (12–18 years old) [176]. It is suggested that carriers of the *DRD4* (rs1800955) C allele could have reduced concussion susceptibility via a neuro-protective response from greater D4 receptor availability, thus inhibiting risk-taking behaviour.

#### 5.1.10. Serotonin Transporter Polymorphisms

The serotonin transporter gene (*solute carrier family 6 member 4, SLC6A4*) is reported to play a role in personality and behavioural traits [127]. The 5-HTTLPR (rs4795541) polymorphism is a variable number tandem repeat (up to 28 bp) insertion (long (L) allele) or deletion (short (S) allele) located in the promoter region of the 5-HTT-encoding gene *SLC6A4*. Reduced serotonin transporter expression is reported for the S allele (Appendix A) [128]. An additional an A/G SNP (rs25531) within the long allele of rs4795541 appears to modulate serotonin transporter expression further, as the L_G_ allele has been associated with lower serotonin expression than the L_A_ allele [177]. The S allele of rs4795541 has previously been associated with harm avoidance, impulsive behaviours and risk taking, though inconsistently. In 78 sibling pairs, harm avoidance scores were higher for S allele carriers than L allele carriers [129] and individuals possessing the LL genotype have been observed to be more risk taking during decision-making trials [130]. However, children and adolescents carrying the S allele showed more impulsive behaviour such as delay aversion during target-game activity [131]. Recently, it has been observed that 5-HTTLPR low (S_A_/S_A_) and intermediate (S_A_/L_A_, S_A_/L_G_, L_A_/L_G_, L_G_/L_G_)-possessing junior RU players displayed less harm avoidance behaviour [21]. These findings suggest that genetic variants associated with personality and thus behavioural traits could influence concussion risk in rugby.

#### 5.1.11. Endothelial Nitric Oxide Synthase Polymorphism

Nitric oxide (NO) plays a major role in the maintenance of cerebral blood flow and is synthesised by three NO synthase isoforms—endothelial (eNOS), neuronal and inducible [132,133]. Nitric oxide is reduced post-TBI under experimental conditions [134,135] and the *NOS3* -786T/C (rs20707044) promoter polymorphism has been associated with promoter region activity, reduced NO synthesis and cerebral vasospasm [136] (Appendix A).

Robertson et al. [137] reported lower cerebral blood flow in -786 C allele (rs2070744)-carrying patients with severe TBI. Multifactorial pathophysiological mechanisms contribute to the reduction in cerebral blood flow as a result of sustaining a concussion [89]. Thus, it could be postulated that possession of a -786 C allele could negatively affect a concussed individual, due to further reduced cerebral blood flow and this warrants further investigation.

#### 5.1.12. Angiotensin I-Converting Enzyme Variants

Cerebral blood flow and autoregulation can be reduced following TBI [178]. The *angiotensin I-converting enzyme* (*ACE*) (rs4646994) insertion (I)/deletion (D) polymorphism (Appendix A) has been associated with regulating blood pressure and cerebral circulation [138]. The DD genotype is associated with higher ACE activity [138] and the D allele has been associated with worse cognitive and motor outcome one month after moderate–severe TBI [139]. Other *ACE* polymorphisms (rs7221780 and rs8066276) have been associated with worse Glasgow Outcome Scale scores 6 months post-TBI [140].

#### 5.1.13. Tumour Necrosis Factor Polymorphisms

Inflammatory mediator cytokines can play contrasting roles in TBI, as they could exacerbate effects in early phases and could affect recovery and repair in the later phases [87,179]. Immediately post-TBI, proinflammatory cytokine tumour necrosis factors (TNFs) are upregulated and return to baseline levels within 24-h [180]. TNFs mediate neuronal apoptosis in the early phase of TBI and facilitate repair in the long term [141,142]. In patients with moderate–severe TBI, carriers of an A allele at position *TNF*-308 (rs1800629) (Appendix A) had an increased risk of unfavourable outcome six months post-TBI compared to noncarriers [143]. Located in the promoter region of *TNF*, the A allele has been associated with increased gene expression and as a result is postulated to increase risk of unfavourable outcome post-TBI [143,181].

#### 5.1.14. Transforming Growth Factor Beta1 Polymorphism

The suppressive cytokine transforming growth factor beta1 (TGFB1) plays a role in regulating inflammation and is encoded by the *transforming growth factor beta 1* (*TGFB1*) gene [144]. Two polymorphisms within the promoter region of *TGFB1* (-800 G/A rs1800468 and -509 C/T rs1800469) (Appendix A) have been associated with altered TGFB plasma levels [145]. However, Waters et al. [143] reported no association between these *TGFB1* polymorphisms and overall outcome in severe TBI patients.

#### 5.1.15. Interleukin 1 Alpha and Interleukin 1 Beta Polymorphisms

Interleukin 1 alpha (IL1A) and interleukin 1 beta (IL1B) are proinflammatory cytokines (Appendix A). In experimental models, both IL1A and IL1B levels are increased within hours following a TBI and can remain elevated for days [182]. There are inconsistent findings regarding *IL1A* and *IL1B* polymorphisms and outcome post-TBI. The G allele of *IL1B* -511 (rs16944) and the T allele of +3953 (rs1143634) have been associated with a six-month unfavourable outcome in severe TBI patients [146]. However, Waters et al. [143] observed no association between *IL1A* and *IL1B* polymorphisms and a six-month unfavourable outcome. Furthermore, associations with secondary complications such as seizures and raised intracranial pressure have been reported for the T allele of *IL1A* -899 (rs1800587) and the T allele of *IL1B* +3953 (rs1143634).

#### 5.1.16. Interleukin 6 Receptor Polymorphism

Interleukin 6 plays a role in the inflammatory process following injury through both pro- and anti-inflammatory properties [147]. A SNP exists at residue 358 (rs2228145) of the *interleukin 6 receptor (IL6R)* gene (Appendix A), the CC genotype of which has been associated with an increased risk of concussion in college athletes [22]. It is postulated that the CC genotype could increase the early inflammatory response post-concussion and lead to reduced cognition [22]. However, Waters et al. [143] reported no associations between *IL6R* promotor polymorphisms and outcome in severe-TBI patients.

## 6. Conclusions and Future Directions

Elite rugby players are exposed to a higher risk of concussion during a playing career than athletes in many other sports. A critical step in better understanding inter-individual variability in the risk of sustaining a concussion and the duration of recovery following a concussion involves identifying genetic variations associated with those risks. The literature has already identified several genetic factors with inter-individual variability in concussion and TBI incidence, severity and recovery. The genes and polymorphisms reviewed here, along with many others, need to be investigated further in relation to incidence rates and recovery from concussion, particularly in a sport such as rugby with a relatively high concussion risk. The number of individuals competing in truly elite rugby is low, so highly collaborative research is required to achieve sample sizes sufficient for satisfactory statistical power.

The inter-individual variation in outcomes following concussion makes predictions of recovery and future risk difficult. This variability could mean there is a future valuable role for genetic screening of concussion-associated risk polymorphisms to complement other data. Achieving elite status in a sport such as rugby is a multifactorial accomplishment due to the complex interactions of multiple environmental factors and the polygenic nature of inherited characteristics and predispositions. Epigenetic regulation of genome function in the context of particular environmental stimuli might also be important in modulating the risk of concussion injury and the rate of recovery. Elite rugby players are exposed to one of the highest risks of concussion in team sports, so distinctive genetic characteristics may exist in those athletes that offer advantages in resisting frequent or severe concussions, relative to those less successful in the sport. Athletes in other sports with a high risk of concussion are also particularly likely to benefit from this kind of genetic resistance to injury. The findings, however, could be applied to a wider range of sports, including those with a lower but still extant risk of concussion. Thus, future research that combines an individual’s concussion history and other phenotypes with detailed genomic information could facilitate more personalised management of concussion and eventually help protect athletes from unfavourable longer-term health outcomes.

## Figures and Tables

**Figure 1 sports-09-00019-f001:**
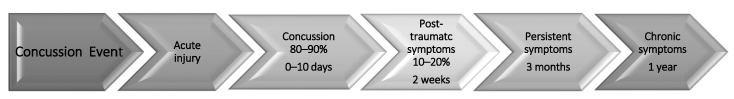
Sequence of events and possible recovery durations post-concussion.

**Figure 2 sports-09-00019-f002:**
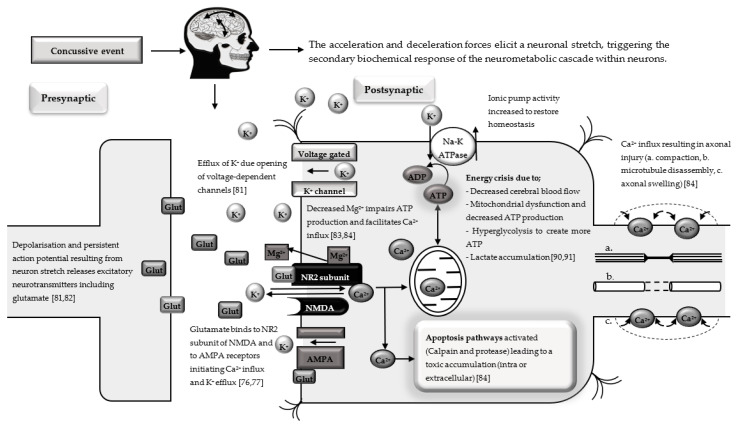
Concussive event leading to the neurometabolic cascade. Glut, glutamate; K^+^, potassium; Ca^2+^, calcium; Mg^2+^, magnesium; AMPA, α -amino-3-hydroxy-5methyl-4-isoxazole-propionic acid.

**Table 1 sports-09-00019-t001:** Candidate genes linked to TBI.

Gene Name	Gene Abbreviation	Polymorphism Identifier	Relevant Effects Associated with TBI
*Apolipoprotein E*	*APOE*	rs429358 rs7412 rs405509	Affects repair and plasticity of the brain [97,98]. APOE isoforms have differing effects on neurite extension, which can influence ability to recover post-concussion [97,98,99,100].Associated with functional regulation of *APOE* transcription [101,102].
*Microtubule-associated protein tau*	*MAPT*	rs10445337 rs2435211 rs2435200	Affects repair and plasticity of the brain via modulation of microtubule formation, structural stabilisation of the neuronal axons and drives growth of neurites [103,104].
*Neurofilament heavy*	*NEFH*	rs165602	Affects repair and plasticity of the brain via modulation of the neuronal cytoskeleton is to resist the resultant strain caused by biomechanical forces [105].
*Membrane metalloendopeptidase*	*MME*	GT repeat promoter polymorphism of neprilysin	Affects repair and plasticity of the brain as this gene encodes for the neprilysin protease which degrades Aβ proteins [106,107,108].
*Brain-derived neurotrophic factor* *antisense RNA*	*BDNF-AS*	rs6265	Affects repair and plasticity of the brain via strengthening existing synaptic connections and modulating the creation of new synapses [109,110,111].
*Glutamate ionotropic receptor NMDA type subunit 2A promoter*	*GRIN2A*	rs3219790	Affects duration of concussion via potential modulation of glutamate-gated ion channel proteins [112,113,114,115].
*Catechol-O-methyltransferase*	*COMT*	rs4680	Affects cognitive behavioural capacity post-concussion and could increase impulsivity and risk taking [116,117,118].
*Ankyrin repeat and kinase domain containing 1*	*ANKK1*	rs1800497	Affects cognitive behavioural capacity via modulation of expression of D2 receptors [119,120,121,122,123,124].
*Dopamine receptor D2* *Dopamine receptor D4*	*DRD2* *DRD4*	rs12364283 rs1076560 rs1800955	Affects personality traits, associated with risk-taking behaviours (impulsivity, behavioural inhibition and novelty seeking) [125,126].
*Solute carrier family 6 member 4*	*SLC6A4*	rs4795541 rs25531	Reported to play a role in personality and behavior via increased harm avoidance and impulsivity behaviours [127,128,129,130,131].
*Endothelial nitric oxide synthase*	*NOS3*	rs2070744	Could affect severity of concussion and cognitive behavioural capacity post-concussion via modulation of cerebral vasospasm [132,133,134,135,136,137].
*Angiotensin I-converting enzyme*	*ACE*	rs4646994 rs7221780 rs8066276	Affects cognitive behavioural capacity post-concussion via modulation of cerebral blood flow [138,139,140].
*Tumour necrosis factor*	*TNF*	rs1800629 rs1800468 rs1800469	Could affect neuroinflammation and severity of concussion [141,142,143].
*Transforming growth factor beta 1*	*TGFB1*	rs1800468 rs1800469	Regulation of the anti-inflammatory mediator TGFB1 could affect severity of concussion [144,145].
*Interleukin 1 alpha* *interleukin 1 beta*	*IL1A* *IL1B*	rs1800587 rs16944 rs1143634	Affects severity of TBI via potential modulation of the inflammatory process and secondary conditions [146].
*Interleukin 6 receptor*	*IL6R*	rs2228145	Affects severity of concussion potential via modulation of the inflammatory process and cognitive behavioural capacity post-concussion [147].

**Table 2 sports-09-00019-t002:** Three isoforms and six possible genotypes of *APOE*.

APOE Isoform	APOE Genotype	rs429358	rs7412
ε2		T	T
ε3		T	C
ε4		C	C
	ε2/ε2	TT	TT
	ε2/ε3	TT	CT
	ε2/ε4	CT	CT
	ε3/ε3	TT	CC
	ε3/ε4	CT	CC
	ε4/ε4	CC	CC

## Data Availability

Data is contained within the article or Appendix A.

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
