# Peer review of "Genetic Factors That Could Affect Concussion Risk in Elite Rugby"

_sports, 2021, doi:10.3390/sports9020019_

Round 1

Reviewer 1 Report

In my opinion, this is a very detailed review about concussion risk in elite rugby. I would suggest the authors add reference PMIDs and reported P-values for the listed SNPs in Table1.

Author Response

I would suggest the authors add reference PMIDs and reported P-values for the listed SNPs in Table1.

Thank you for these suggestions. The citations for relevant literature have now been included in Table 1. However, we prefer not to include the P-values as this table summarises multiple articles containing multiple p-values, so we feel that including them would make the concise summary less clear to readers.

Reviewer 2 Report

This is a very interesting review paper focused on the role genetics play in concussion incidence and post-concussion symptoms.

The title of your paper specifies elite rugby players. However, the paper as a whole is primarily generalized to concussion (not sport specific). In my opinion "in Elite Rugby" should be removed from the title and the few paragraphs in your paper focusing on rugby should be generalized to concussion.

Minor Edits

Introduction

Lines 52-62: This paragraph can be deleted. Although this list of facts is interesting and important to understanding concussion, it strays from the focus of this review and therefore is not needed.

Line 80: delete “however” or change it to a different word.

Line 92: delete “currently” in the second sentence.

Incidence rate and severity of concussion in rugby

Line 158: remove the superscript from the references.

Pathophysiology of concussion

Line 216: add a space after the period.

Genetic associations with concussion

Order paragraphs to match the order they are listed in Table 1

Line 319: define what “OR” stands for.

Line 411: Remove space between “alleles” and “associated”

Figures

The words within figure 2 are difficult to read. Please improve figure quality.

In figure 2 the sentence between the two neurons has the word receptors and receptor after AMPA. Delete one of them please.

Tables

Table 1: “Apolipoprotein E” should be move up a bit to line up with APOE

Table 1: The way the table is formatted it appears that Dopamine receptor D4 does not have a description for the “relevant effects associated with TBI”. If the relevant effects are the same as D3 please move the description down a little so the text lines up with both

Table 1: For tumor necrosis factor, what is meant by “the worse outcome”? Please clarify or choose more descriptive wording

Table 1: Move the relevant effects associated with TBI for interleukin 1 alpha and beta down just a bit so it lines up better with all the polymorphisms.

Table 1: Add “via” between “potential” and “modulation” in the relevant effects associated with TBI for interleukin 6 receptor.

Table 1: Please check alignment for all categories

Table 2: Adjust 1st column so the titles “isoform” and “genotypes” fit on one line.

Table 3: Table 3 is mentioned in the text but there is no table labeled “Table 3”

References

Line 537: The first reference is out of alignment with the rest

Author Response

The title of your paper specifies elite rugby players. However, the paper as a whole is primarily generalized to concussion (not sport specific). In my opinion "in Elite Rugby" should be removed from the title and the few paragraphs in your paper focusing on rugby should be generalized to concussion.

Thank you for this suggestion. However, we believe that keeping ‘in Elite Rugby’ in the title and keeping the focused paragraph on rugby would be more appropriate due to the recent broad interest and discussion of concussion in elite rugby, and the fact that rugby is a high-risk concussion environment. Indeed, genetic factors are more likely to play a role where the environmental risk is high, as opposed to lower risk sports and the general population who would arguably benefit more from concussion prevention strategies, which are out of the scope of this review. However, prompted by your comments and those of Reviewer 3, we have added text to the conclusion that refers to other sports.

Minor Edits

Introduction

Lines 52-62: This paragraph can be deleted. Although this list of facts is interesting and important to understanding concussion, it strays from the focus of this review and therefore is not needed.

Thank you for this suggestion. The paragraph has been deleted but we have retained a little of the text and merged it with the next paragraph to avoid the impression that genetics is the only relevant risk factor.

Line 80: delete “however” or change it to a different word.

Thank you for this suggestion. The word ‘however’ has been deleted.

Line 92: delete “currently” in the second sentence.

Thank you for this suggestion. The word ‘currently’ has been deleted.

Incidence rate and severity of concussion in rugby

Line 158: remove the superscript from the references.-

Thank you for pointing this out. Superscript have been removed from the references

Pathophysiology of concussion

Line 216: add a space after the period.

Thank you for pointing this out. A space after the period has been added

Genetic associations with concussion

Order paragraphs to match the order they are listed in Table 1

Thank you for pointing this out. The paragraphs have been arranged in the same order as the table.

Line 319: define what “OR” stands for.

Thank you for pointing this out. “odds ratio” has been written in full instead of the abbreviation. 

Line 411: Remove space between “alleles” and “associated”

Thank you for pointing this out. The space has been removed.

Figures

The words within figure 2 are difficult to read. Please improve figure quality.

Thank you for pointing this out. Image quality and font size has been amended in figure 2.

In figure 2 the sentence between the two neurons has the word receptors and receptor after AMPA. Delete one of them please.

Thank you for pointing this out. Receptor has been deleted in figure 2.

Tables

Table 1: “Apolipoprotein E” should be move up a bit to line up with APOE.

Thank you for pointing this out. Alignment of APOE has been adjusted in Table 1.

Table 1: The way the table is formatted it appears that Dopamine receptor D4 does not have a description for the “relevant effects associated with TBI”. If the relevant effects are the same as D3 please move the description down a little so the text lines up with both.

Thank you for pointing this out. Alignment of D3 and D4 description has been adjusted so it appears both variants have the same description, in Table 1.

Table 1: For tumor necrosis factor, what is meant by “the worse outcome”? Please clarify or choose more descriptive wording.

Thank you for pointing this out. Description has been changed to ‘Could affect neuroinflammation and severity of concussion’ in Table 1.

Table 1: Move the relevant effects associated with TBI for interleukin 1 alpha and beta down just a bit so it lines up better with all the polymorphisms.

Thank you for pointing this out. Alignment of effects associated with TBI for interleukin 1 alpha and beta has been adjusted in Table 1.

Table 1: Add “via” between “potential” and “modulation” in the relevant effects associated with TBI for interleukin 6 receptor.

Thank you for pointing this out. Via has been added in Table 1.

Table 1: Please check alignment for all categories-

Thank you for pointing this out. Table has been left aligned in Table 1.

Table 2: Adjust 1st column so the titles “isoform” and “genotypes” fit on one line.

Thank you for this suggestion. We have adjusted the table as requested.

Table 3: Table 3 is mentioned in the text but there is no table labeled “Table 3”.

Thank you for pointing this out. The mention of Table 3 has been removed

References

Line 537: The first reference is out of alignment with the rest.

Thank you for pointing this out. The first reference has been aligned with the rest.

Reviewer 3 Report

In the manuscript ‘Genetic factors that could affect concussion risk in elite rugby’, Antrobus et al have described in detail the prevalence, mechanisms, and pathophysiology of concussion incidents in professional rugby players. The authors further propose to investigate candidate genes for variants that might be linked to differences in traumatic brain injury incidence and severity between individuals.

Overall the manuscript is well written and makes a good case to focus on the association between injury outcomes and genetics in Rugby.

Minor points:

  1. Although the review is focused on the impact of concussion on elite rugby players, the authors could briefly discuss how findings from the proposed studies could find widespread applications in other contact sports and in the general population susceptible to injuries and concussions.
  2. Figure 2 could have a better resolution and a bigger font size. It is difficult to read the information within the figure.

Author Response

  1. Although the review is focused on the impact of concussion on elite rugby players, the authors could briefly discuss how findings from the proposed studies could find widespread applications in other contact sports and in the general population susceptible to injuries and concussions.

Thank you for this suggestion. Because our review is focussed on sport participants who, even if professional, have to a greater or lesser extent volunteered to place themselves in a high-risk environment, we do not feel that referring to the general population (non-athletes) in our review would be appropriate. However, prompted by your comment and those of Reviewer 2, we have added text to the conclusion that now refers to other sports, including those with relatively high and relatively low risks of concussion.

  1. Figure 2 could have a better resolution and a bigger font size. It is difficult to read the information within the figure.

Thank you for pointing this out. Image quality and font size has been amended in figure

Round 2

Reviewer 2 Report

Thank you for addressing all my comments and explaining why you want to keep the focus on elite rugby.